# Genome-wide perturbations of Alu expression and Alu-associated post-transcriptional regulations distinguish oligodendroglioma from other gliomas

Taeyoung Hwang [1✉], Sojin Kim[2], Tamrin Chowdhury [2], Hyeon Jong Yu[2], Kyung-Min Kim[2], Ho Kang[2], Jae-Kyung Won[3], Sung-Hye Park [3], Joo Heon Shin[1] & Chul-Kee Park [2✉]

*Alu* is a primate-specific repeat element in the human genome and has been increasingly appreciated as a regulatory element in many biological processes. But the appreciation of *Alu* has been limited in tumorigenesis, especially for brain tumor. To investigate the relevance of *Alu* to the gliomagenesis, we studied *Alu* element-associated post-transcriptional processes and the RNA expression of the element by performing RNA-seq for a total of 41 pairs of neurotypical and diverse glioma brain tissues. We find that A-to-I editing and circular RNA levels, as well as *Alu* RNA expression, are decreased overall in gliomas, compared to normal tissue. Interestingly, grade 2 oligodendrogliomas are least affected in A-to-I editing and circular RNA levels among gliomas, whereas they have a higher proportion of down-regulated *Alu* subfamilies, compared to the other gliomas. These findings collectively imply a unique pattern of *Alu*-associated transcriptomes in grade 2 oligodendroglioma, providing an insight to gliomagenesis from the perspective of an evolutionary genetic element.

[1] Lieber Institute for Brain Development, Johns Hopkins Medical Campus, Baltimore, MD 21205, USA. [2] Department of Neurosurgery, Seoul National University Hospital, Seoul National University College of Medicine, Seoul 03080, Republic of Korea. [3] Department of Pathology, Seoul National University Hospital, Seoul National University College of Medicine, Seoul 03080, Republic of Korea. ✉email: taeyoung.hwang@libd.org; nsckpark@snu.ac.kr

A large proportion of the human genome consists of repetitive elements. Transposable elements (TEs), a major repeat element, account for at least 45% of the human genome, while coding sequences comprises less than 3%[1]. Among TEs, *Alu* is the most abundant repeat element, consisting of about 10% of the human genome[1]. *Alu* is primate-specific and expands through retrotransposition, an amplifying process of TE through an RNA intermediate. *Alu* is transcribed mainly by RNA polymerase III[2,3], generating an about 300 nucleotides (nts)-long noncoding RNA (ncRNA). Alu RNA tends to form double-stranded RNA (dsRNA) as it has two monomers facing each other.

Although *Alu* was considered as junk DNA in the past, there has emerged increasing evidence that this element plays important regulatory roles in diverse cellular processes[4]. Specifically, *Alu*'s regulatory roles are manifest at DNA, RNA, and post-transcriptional levels. At the DNA level, *Alu* insertion can generate regulatory elements, including alternative splicing[5] and enhancer function[3,6]. The frequent location of *Alu* in genic regions exert its effect at post-transcriptional regulations including A-to-I editing[7–9], an RNA modification changing RNA sequence at a single nucleotide from adenosine to inosine, and formation of circular RNA[10], a single-stranded RNA with a covalently closed loop structure. In particular, it has been reported that *Alu*-mediated A-to-I editing is tightly regulated[11] with a potential to contribute to expand a repertoire of RNA[12]. The dsRNA structure of *Alu* is known to facilitates A-to-I editing and circular RNA formation[8,13]. *Alu* also harbors a nuclear localization signal of long noncoding RNA (lncRNA)[14]. As a short ncRNA, *Alu* RNA affects transcription by modulating RNA polymerase II (Pol II)[15,16]. In addition, *Alu* RNA has been reported to influence translation[17].

*Alu* also has implications in human diseases including cancer and neuropathological disorders. In cancer biology, *Alu* RNA has been studied in hepatocellular carcinoma[18] and in a metastatic colorectal cancer cell line[19], both showing that increased levels of *Alu* RNA are associated with tumor development. The dysregulation of *Alu*-mediated post-transcriptional processes has been cited in the context of neuropathological diseases[20,21]. *Alu*-associated A-to-I editing and circular RNA are abundant in neuronal tissues and involved in neuronal differentiation and potentially in developmental disorders[22,23]. In addition, *Alu* has been hypothesized to potentially play a role in neurodegenerative diseases[20,24]. However, the comprehensive study of *Alu* in the context of brain tumor is limited although previous studies showed that A-to-I editing dysregulation is involved in brain tumors[25,26]. This is due at least in part to an underappreciation of the possible role of evolutionary and developmental processes on tumor pathogenesis and the difficulty in studying *Alu*'s noncoding functions in biological systems.

We have previously shown that the *Alu*-associated A-to-I editing pattern of glioblastoma (GBM) is similar to that of early-stage neurodevelopment[22]. In the present study, we extended our analysis to multiple types of gliomas to test whether *Alu*-associated dynamic processes at both post-transcriptional and RNA levels are related to the progression of glioma. Using a spectrum of glioma samples with up-to-date molecular classification[27,28], we investigated the expression levels of *Alu* RNA and the two *Alu*-associated post-transcriptional regulations that are abundant in human brain tissue, i.e., A-to-I editing and circular RNA expression. Our analysis first confirmed a previous finding that A-to-I editing sites and circular RNAs are significantly associated with *Alu* element in brain tissues. By comparing genome-wide patterns between matched tumor and neurotypical brain samples, we observed that A-to-I editing levels and circular RNA expression are perturbed in gliomas and are globally decreased in high-grade gliomas. In particular, grade 2 oligodendroglioma, a glioma of favorable prognosis, present the least global decrease in both A-to-I editing levels and circular RNA expression. Also, a unique pattern in *Alu* RNA expression was found in grade 2 oligodendroglioma, specifically wide downregulation of *Alu* subfamilies in these tumors relative to matched normal brain. Finally, we found that Adenosine deaminase RNA specific B1 (*ADARB1*, also known as *ADAR2*) was downregulated in gliomas other than grade 2 oligodendrogliomas, potentially contributing to the observed decrease in A-to-I editing and circular RNA formation. Our results demonstrate *Alu* is associated with gliomas through its own expression and associated post-transcriptional regulations, providing a potential insight into the molecular mechanisms of gliomas from the perspective of a primate-specific repetitive element.

## Results

### Catalogue and pattern of RNA editing in glioma detected by strand-specific RNA-seq.

We performed strand-specific RNA-seq for tumor and matched normal samples obtained from 41 patients across various pathologies of gliomas spanning *IDH* mutant and 1p/19q-codeleted oligodendroglioma grades 2 and 3 (O2 and O3), *IDH* mutant astrocytoma grades 2, 3 and 4 (A2, A3, and A4), and GBM (Supplementary Table 1). In addition, we developed a computational pipeline to identify RNA-editing sites in a genome-wide context from strand-specific RNA-seq (see Methods in detail). While many previous methods identifying RNA-editing sites from RNA-seq depend on precompiled gene annotation to assign RNA editing types[22,29–31], our pipeline considers strand information embedded in stranded RNA-seq to determine RNA-editing types, allowing unbiased identification of RNA-editing sites. After we applied the computational pipeline to individual samples in our dataset, we additionally filtered potential DNA variants that have inconsistency across the samples in terms of editing type or strand. Also, we only chose the sites that were found in at least two patients in order to minimize the contamination of rare DNA variants in our list of RNA-editing sites. As a result, we identified a total of 700,471 RNA variant sites across samples in our dataset. The RNA variants were predominated by A-to-G which is a representation of A-to-I editing in RNA-seq (Fig. 1a), comprising about 81% of RNA variant sites (number: 572,385). The second dominant type was C-to-U whose proportion is about 4%. This type of editing is known to be found in human cells[32,33]. If we consider potential technical errors in strand-specific RNA-seq, A-to-I and C-to-U editing can be identified as their reverse complemented forms of T-to-C and G-to-A, respectively. These four types accounted for about 93% of all the identified sites, indicating our pipeline's higher specificity of identification.

As expected, most A-to-I editing sites in our list show their strong association with *Alu*: about 82% of the sites were identified in annotated *Alu* element regions. Also, most A-to-I editing sites were found in intronic regions (93%). When we compared our list of A-to-I editing sites with a public database of A-to-I editing sites (REDIportal)[34], 88% of our sites were found in the database. The numbers of A-to-I editing sites normalized by sequencing depth vary across patients and tend to be smaller in tumor compared normal tissues (Fig. 1b, $N = 41$ patients, two-sided paired *t*-test of log (base:10) transformation of the normalized number of A-to-I editing sites: *t*-statistic = 3.29, *p* value = 0.002076, two-sided paired Wilcoxon signed rank test: *p* value = 0.001076). Many A-to-I editing sites were not found commonly across the patients. The proportion of A-to-I editing sites that were shared by patients decreases according to the increasing number of patients observing the sites (Supplementary Fig. 1).

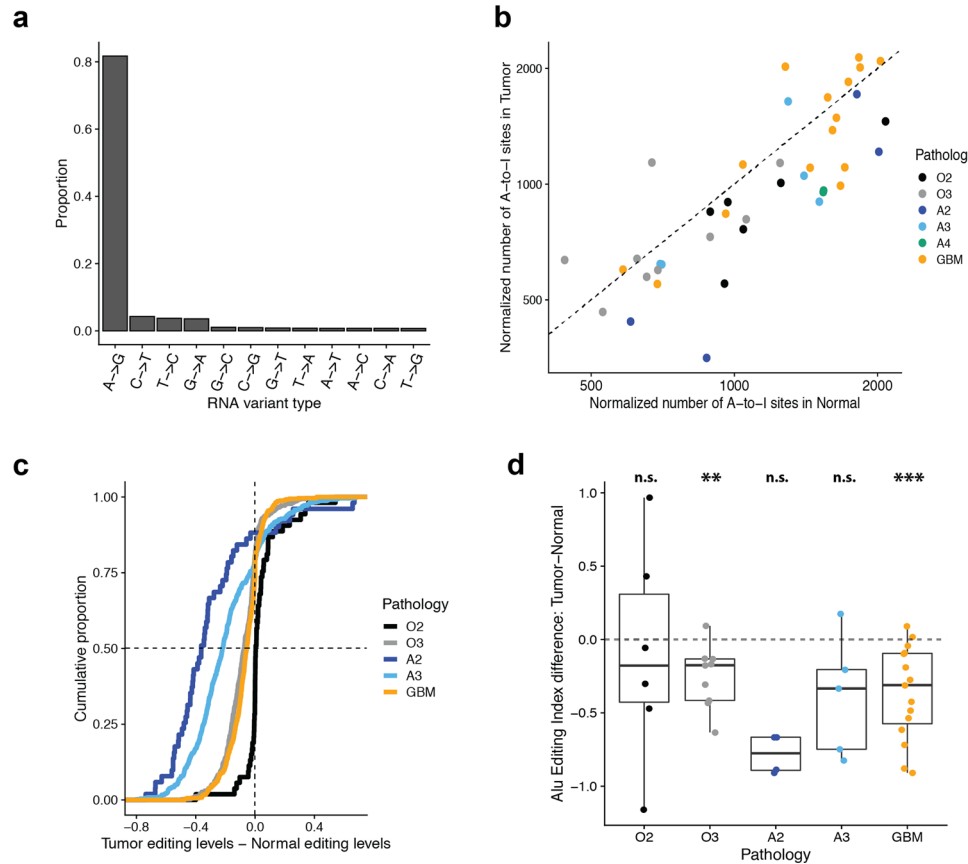

**Fig. 1 RNA-editing sites and A-to-I editing levels in glioma. a** Our unbiased computational pipeline detects RNA variants in 82 samples from patients of brain tumors. A-to-G type is most abundant among the identified RNA variants, indicating that most RNA variants are A-to-I editing sites. **b** Numbers of A-to-I editing sites per one million reads were plotted for individual patients (dots) of different pathologies (colors) according to tumor and normal tissues (y- and x-axis values, respectively): grade 2 oligodendroglioma *IDH* mutant and 1p/19q-codeleted (O2), grade 3 oligodendroglioma *IDH* mutant and 1p/19q-codeleted (O3), grade 2 *IDH* mutant astrocytoma (A2), grade 3 *IDH* mutant astrocytoma (A3), grade 4 *IDH* mutant astrocytoma (A4), and glioblastoma (GBM). The dashed line indicates the line of y = x. **c** Distributions of A-to-I editing level changes in tumors relative to matched normal tissues were described in a cumulative way for different pathologies: the shift of the curve to the left to 0 means overall decrease in tumor relative to normal tissues. **d** Distributions of Alu Editing Index (AEI) differences per pathology: each dot is a patient (N = 6, 9, 4, 5, 15 patients for O2, O3, A2, A3, GBM, respectively). The *p* values testing whether means are different from 0 by Wilcoxon text (paired), were shown (\*\*: *p* value < 0.01, \*\*\*: *p* value < 0.001, ns not significant, *p* value > 0.05). In a boxplot, the whiskers extend from the bottom and top of the box (the first and third quartiles) to the largest and the smallest value no further than 1.5 \* interquartile range.

But we also identified that some numbers (3415) of A-to-I editing sites were found in all the patients and they had a potential to cluster samples into tumor and normal tissues (Supplementary Fig. 2 and Supplementary Data 1), which implies that these A-to-I editing sites are regulated in human brain tissues, despite the variability of individual samples.

In order to identify A-to-I editing sites that show different editing levels between tumor and normal tissues, we performed regression-based statistical tests comparing tumor and normal tissues, while controlling patient-specific differences (see Methods). As pathology is a clear factor contributing to variation in our dataset (Supplementary Fig. 3), we conducted the statistical tests per a pathology, except for *IDH* mutant astrocytoma grade 4 (A4) due to the small number of patients (*n* = 2). We first found that the differential sites were enriched with the commonly found sites: 49.0% of 1360 differential sites in any pathology are shared by all the patients while 0.5% of nondifferential sites were such sites. Per pathology, we found that about 0.3~7.7% of A-to-I editing sites showed differential editing between tumor and matched normal tissues (Table 1 and Supplementary Data 2). Interestingly, low grade gliomas, O2 and A2 showed relatively lower number of differential editing sites (0.4% for O2 and 0.3%

for A2) compared with high-grade gliomas (6.5%, 3.3%, and 7.7% for O3, A3, and GBM, respectively). When we checked the overlap of the differentially-edited sites between two pathologies, we found that the degree of overlap between O2 and each of the others was much smaller than comparisons between other pairs of pathologies (Supplementary Figure 4).

We also compared the overall distribution of A-to-I editing level differences between tumor and matched normal tissues. In Fig. 1c, the changes of A-to-I editing levels are summarized by a cumulative distribution whose shift to the left to the zero indicates the overall decrease of A-to-I editing levels in tumor relative to matched normal tissues. We found that most A-to-I editing sites in higher grades gliomas or nonoligodendroglioma were decreased in general in tumors relative to the matched normal tissues. In contrast, grade 2 oligodendroglioma showed no shift or no bias in difference of A-to-I editing levels. Specifically, average A-to-I editing levels do not show significant differences between tumors and normal tissues in O2 (*N* = 53 sites, average difference = 0.04, two-sided *t* test *p* value = 0.05665, two-sided Wilcoxon signed rank test *p* value = 0.02186), while the others have significant decreases: O3 (*N* = 496 sites, average difference = −0.08, two-sided *t* test *p* value < $10^{-15}$, two-sided Wilcoxon

**Table 1 Summary of A-to-I editing sites according to pathologies in glioma Test sites are the A-to-I editing sites found in all patients for a given pathology.**

| Pathology | Number of patients | Number of test sites | Number of significant sites | Proportion of significant sites (%) |
|---|---|---|---|---|
| O2 | 6 | 13,856 | 53 | 0.4 |
| O3 | 9 | 7651 | 496 | 6.5 |
| A2 | 4 | 15,860 | 51 | 0.3 |
| A3 | 5 | 14,153 | 473 | 3.4 |
| GBM | 15 | 6604 | 507 | 7.7 |

Significant sites were determined by FDR-adjusted $p$ value cutoff 0.05 in the statistical comparisons of A-to-I editing levels between tumor and matched normal tissues.

signed rank test $p$ value $< 10^{-15}$), A2 ($N = 51$ sites, average difference $= -0.30$, two-sided $t$ test $p$ value $< 10^{-8}$, two-sided Wilcoxon signed rank test $p$ value $< 10^{-6}$), A3 ($N = 473$ sites, average difference $= -0.19$, two-sided $t$ test $p$ value $< 10^{-15}$, two-sided Wilcoxon signed rank test $p$ value $< 10^{-15}$), GBM ($N = 507$ sites, average difference $= -0.07$, two-sided $t$ test $p$ value $< 10^{-15}$, two-sided Wilcoxon signed rank test $p$ value $< 10^{-15}$). We further observed that the increasing grade in astrocytoma leads to less decrease in average editing levels while oligodendroglioma showed an opposite direction on average. But this trend might be due to differing sample sizes per pathology and should be interpreted cautiously. The loss of A-to-I editing in tumor relative to normal tissues were also confirmed by the Alu editing index (AEI) that was previously proposed to quantify A-to-I editing levels in Alu region[35] (Fig. 1d and Supplementary Fig. 5). All together, these results suggest that both the perturbed sites and the direction of editing level changes are different between grade 2 oligodendroglioma and the other gliomas.

**Expression of circular RNA in glioma.** Another *Alu*-associated post-transcriptional regulation that is abundant in brain tissue is a back-splicing process, where a 5′ splice donor joins an upstream 3′ splice acceptor, generating circular RNA. We identified the genes that produce circular RNA by checking whether a gene has back-splicing RNA-seq junction reads that are signatures of circular RNA (see methods). About 13.5% of annotated genes (7893 among 58,288) have circular RNA signatures in their gene bodies in our brain tissues. As expected, they are significantly associated with *Alu* (Fig. 2a): the group of genes with circular RNA signatures shows 98.5% association with *Alu* element while the group of genes without circular RNA-supporting read only has 36.3% association ($N = 58,288$ genes, fisher exact test $p$ value $< 10^{-15}$). The global level of circular RNA, measured by RNA-seq as the total number of back-splicing junction reads normalized by sequencing depth, was in general smaller in tumors compared to the matched normal tissues (Fig. 2b, $N = 41$ patients, two-sided paired t-test of log-transformed (base:10) numbers: $t$-statistic $= 6.96$, $p$ value $= 2.12*10^{-8}$, two-sided paired Wilcoxon signed rank test: $p$ value $= 2.87*10^{-8}$). We also compared the circular RNA expression rates between tumor and matched normal tissues for a gene per pathology, which are defined by the ratio of the number of RNA-seq back-splicing junction reads relative to the total RNA-seq junction read counts (see methods). We found that 973 genes showed differential circular RNA expression rates in our glioma tissues (Supplementary Data 3). Overall, O2 has the least affected distribution of differential expression rates while the others have variable degrees of decreased expression rates of circular RNA in tumors relative to normal tissues (Fig. 2c): O2 ($N = 256$ genes, average difference: $-0.01$, two-sided paired $t$-test $p$ value $< 10^{-15}$, two-sided Wilcoxon signed rank test $p$ value $< 10^{-15}$), O3 ($N = 567$ genes, average difference: $-0.02$, two-sided paired $t$-test $p$ value $< 10^{-15}$, two-sided Wilcoxon signed rank test $p$ value $< 10^{-15}$), A2 ($N = 371$ genes, average difference: $-0.04$,

two-sided paired t-test $p$ value $< 10^{-15}$, two-sided Wilcoxon signed rank test $p$ value $< 10^{-15}$), A3 ($N = 229$ genes, average difference: $-0.04$, two-sided paired $t$-test $p$ value $< 10^{-15}$, two-sided Wilcoxon signed rank test $p$ value $< 10^{-15}$), GBM ($N = 42$ genes, average difference: $-0.03$, two-sided paired t-test $p$ value $= 4.30*10^{-6}$, two-sided Wilcoxon signed rank test $p$ value $= 4.55*10^{-13}$).

**Expression of *Alu* RNA in glioma.** *Alu* elements themselves can be expressed and affect cellular processes. We estimated the expression levels of annotated 47 *Alu* subfamilies in our samples and compared their expression levels between tumor and the matched normal tissues using the computational pipeline that can handle repeat elements for differential expression analysis (see Methods). In O2, a higher proportion of *Alu* RNAs (37 out of 47 subfamilies) was perturbed in tumors relative to matched normal tissues while the other gliomas only had one to three *Alu* RNAs as significantly changed ones (Fig. 3a). *Alu* RNAs in O2 were also downregulated with other transposable elements (Fig. 3b). In addition, O2 was identified as the most affected tumor type in terms of the overall expression change of *Alu* RNAs (Fig. 3c): average fold changes of tumor relative to normal tissues for O2, O3, A2, A3, A4, and GBM are $-0.52$, $-0.28$, $-0.34$, $-0.23$, $-0.50$, and $-0.16$, respectively in log scale (base:2). All of the pathologies showed statistically significant decreases according to both parametric and nonparametric statistical tests ($N = 47$ subfamilies, two-sided $t$ tests $p$ value $< 10^{-6}$ for all cases, Wilcoxon signed rank test $p$ value $< 10^{-10}$ for all cases).

**Towards an integrative understanding of *Alu*-associated molecular processes in glioma.** In order to understand the decreasing pattern of A-to-I editing and circular RNA expression in gliomas relative to normal tissues, we checked the RNA expression levels of ADAR (Adenosine Deaminase Acting on Rna) families that are known to generate A-to-I editing (Fig. 4a). *ADAR2* mRNA was downregulated significantly in all gliomas except for O2, which is consistent with the identified decreasing patterns of A-to-I editing. In contrast, *ADAR (Adenosine Deaminase Rna Specific)*, also known as *ADAR1*, did not show any significant mRNA level change across gliomas. ADARB2 (Adenosine Deaminase Rna Specific B2), also known as ADAR3, known to antagonize ADAR1 and ADAR2 by competing with them[36,37] also showed some decrease in high-grade gliomas. Therefore, ADAR2 among ADAR families seems to be a *trans* factor underlying the decreased patterns of A-to-I editing that we observed.

*Alu* RNA is known to be transcribed mainly by RNA polymerase III. Although we found that a RNA polymerase III subunit, *POLR3A* expressions tended to be downregulated in some gliomas suggesting a potential contribution of the downregulation of RNA polymerase III on the lower expression of *Alu* RNA in gliomas, statistical significances were only attained in A2 and A4 gliomas (Supplementary Fig. 6).

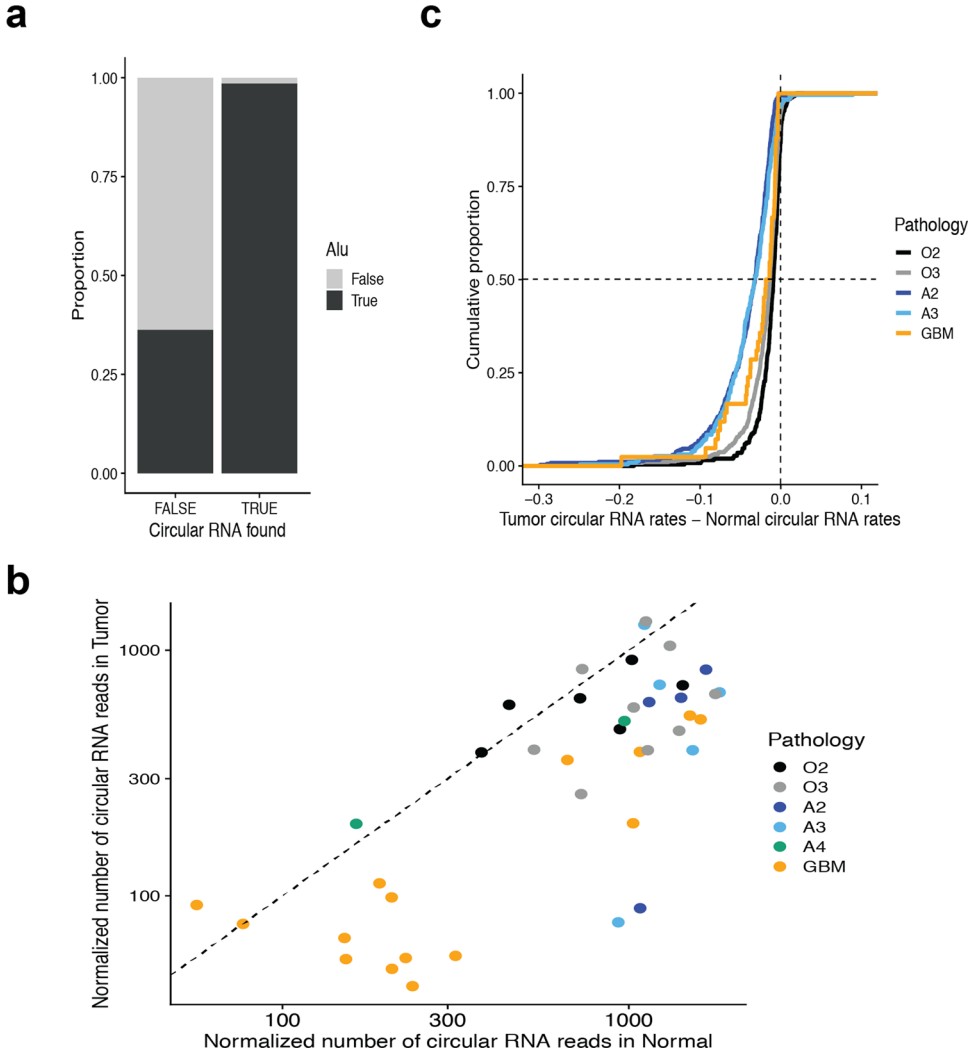

**Fig. 2 Circular RNA expression in glioma. a** Differences of Alu association with gene bodies between the two gene groups with and without circular RNA-supporting RNA-seq reads (back-splicing junction reads). **b** The number of circular RNA-supporting reads per one million reads in RNA-seq. The numbers were plotted for individual patients (dots) and tumor and normal tissues (y and x axis values respectively) according to different pathologies (colors): grade 2 oligodendroglioma *IDH* mutant and 1p/19q-codeleted (O2), grade 3 oligodendroglioma *IDH* mutant and 1p/19q-codeleted (O3), grade 2 *IDH* mutant astrocytoma (A2), grade 3 *IDH* mutant astrocytoma (A3), and glioblastoma (GBM). **c** Distributions of circular RNA expression rate changes in tumors relative to matched normal tissues were described in a cumulative way for different pathologies: the shift of the curve to the left to 0 means overall decrease in tumor relative to normal tissues.

We also sought to understand the relationship between A-to-I editing and circular RNA as they can compete or cooperate for *Alu* element-associated factors such as ADAR. We first checked whether A-to-I editing and circular RNA occurred in the same genes regardless of pathologies, and found that the overlap size (91 genes) is moderate but statistically significant (Fisher exact test, $N = 22725$ genes that are detected by RNA-seq in our samples, $p$ value $< 10^{-15}$) (Fig. 4b). We also tested whether Alu-associated A-to-I editing sites are located in the flanking introns of circular RNAs, which may affect the formation of circular RNA[38]. We found that in 73% (66 genes) among the 91 genes, the perturbed A-to-I editing sites were observed in the flanking introns of any circular RNAs in the same genes. Second, A-to-I editing and circular RNA were compared in terms of the direction of changes in tumor relative to normal tissue. Specifically, we compared the overall A-to-I editing level changes between the two groups of genes defined by whether a gene shows significant circular RNA expression rate changes between tumor and normal. We did not find the notable

consistency between the two processes as the genes with decreased circular RNA expression rates in tumor compared to normal had little decrease in A-to-I levels (Fig. 4c). Finally, we looked into whether the two post-transcriptional processes affect similar pathways by performing gene ontology analyses for the genes harboring differentially-edited sites or the genes showing the different circular RNA expression rates. We found that the genes with perturbed A-to-I editing were over-represented in multiple gene ontology terms (Supplementary Data 4). About 17% of the over-represented terms were shared by at least two pathologies and included the pathways known in neuronal tissues, for examples, glutamate ion channels and the regulation of synapses (Fig. 4d). For circular RNA, we found that different pathways are affected, including chromatin organization and neuron development (Supplementary Figure 7 and Supplementary Data 5). Therefore, our results demonstrated that A-to-I editing and circular RNA perturbations occurs concurrently at some genes, but they affect different genes in different pathways in general.

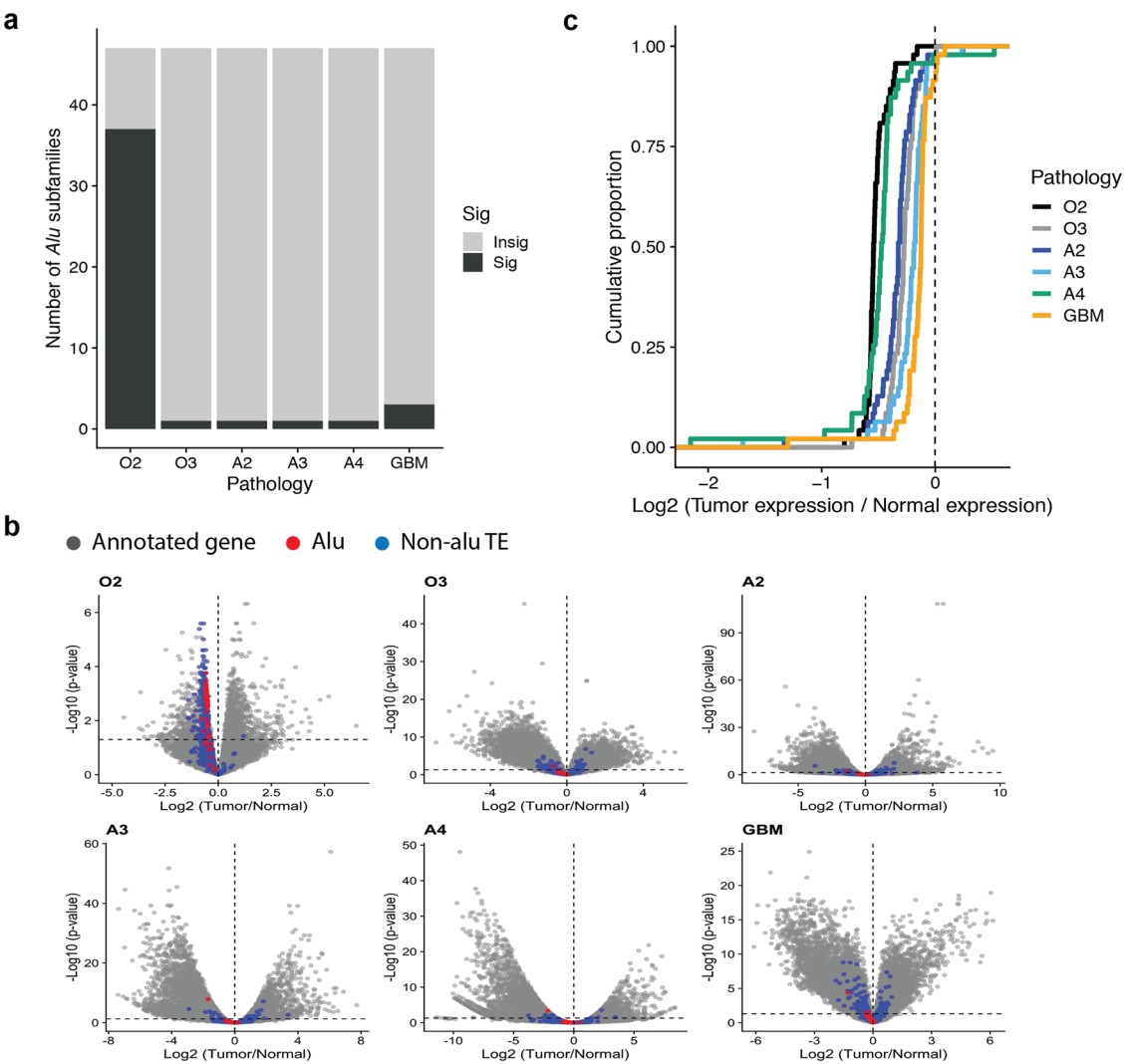

**Fig. 3 Alu RNA expression in glioma. a** The numbers of significantly-perturbed *Alu* RNAs among total 47 Alu subfamilies annotated in GENCODE (v27) for different pathologies. **b** Volcano plots of differential gene expression analyses per brain tumor pathologies. *Alu* elements (red), non-*Alu* transposable elements (blue) and the annotated genes (gray) were described according to the results of differential gene expression analysis. X-axis is fold change of RNA levels in log2 scale (tumor relative to normal) and y-axis is *p* value in the negative log scale (base:10). **c** Distribution of *Alu* RNA expression changes in tumors relative to matched normal tissues were described in a cumulative way for different pathologies. Label of pathologies: grade 2 oligodendroglioma (O2), grade 3 oligodendroglioma (O3), grade 2 astrocytoma (A2), grade 3 astrocytoma (A3), grade 4 astrocytoma (A4), and glioblastoma (GBM).

## Discussion

There have been many attempts to understand neuropathological disorders from the perspective of genome evolution[24,39]. These efforts provide interesting hypotheses relating primate-specific genes or genetic elements to neuropsychiatric disorders as well as neurodegenerative disorders. However, brain cancer has not been appreciated well in terms of primate-specific elements as cancer is generally understood as diseases caused by a genetic mutation associated with oncogenes or tumor suppressor genes and with environmental carcinogens. But increasing evidence suggests that evolutionary mechanisms affect brain tumors[40]. In this study, we looked into a primate-specific *Alu* element to compare various pathologies in gliomas. We used the most recent version of classification criteria of gliomas and matched neurotypical samples for an elaborate comparison between tumor and normal brain tissues. We also performed extensive computational analyses with strand-specific RNA-seq in order to explore *Alu*'s dynamic effects on tumorigenesis in human brain tissue. We found that Alu-associated molecular processes, including Alu RNA expression, A-to-I editing, and circular RNA formation are

perturbed in gliomas, despite the possibility that we might not capture moderate perturbations due to the limited number of patients in each glioma.

An obvious dynamic mode of *Alu* element is its expression as RNA. Although *Alu* is usually suppressed by epigenetic mechanisms, *Alu* RNA is mainly transcribed by RNA polymerase III and dynamically regulated during development and in various diseases[2,41]. However, few studies have been conducted so far on Alu RNA in brain tumor, partly due to difficulty in measurements of repeated sequences in the human genome. We used a rigorous computational pipeline that considers repeat features in counting RNA-seq sequencing reads and controls patient-specific effects in a statistical comparison of Alu RNA expression between tumor and normal tissues. We found downregulation of Alu RNA in almost all pathologies of gliomas at varying degrees. The largest downregulation is observed in grade 2 oligodendroglioma while GBM shows the least downregulation. This perturbation of Alu RNA may contribute to different prognosis between O2 and GBM as a previous study showed that higher levels of Alu RNAs induces epithelial-to-mesenchymal transition

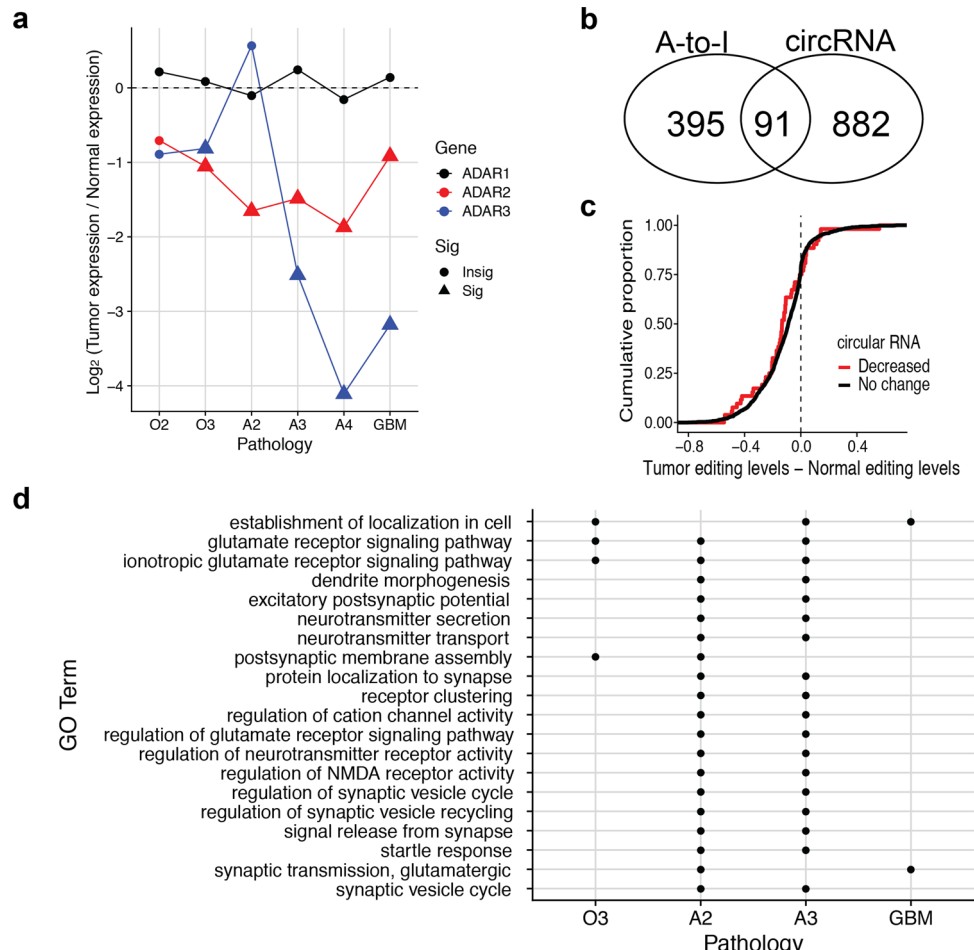

**Fig. 4 Integrative understanding of *Alu*-associated molecular processes in glioma. a** Average fold change of mRNA expression levels of A-to-I editing enzyme ADAR families (*ADAR1*, *ADAR2*, and *ADAR3*). Statistical significance of differential expression was determined by a R package DESeq2: significant if FDR-adjusted *p* value < 0.05 (N = 12, 18, 8, 10, 4, 30 tissues for O2, O3, A2, A3, A4, GBM, respectively). **b** Venn diagram of the genes with differentially-edited A-to-I editing sites and the genes showing differential expression rates of circular RNA. **c** The distributions of A-to-I editing level changes in tumors relative to matched normal tissues were compared between the two gene groups made by whether genes show the decreased circular RNA expression rates in tumors relative to matched normal tissues (two-sided Kolmogorov-Smirnov test *p* value: 0.07202, N = 1578 A-to-I editing sites). **d** Gene ontology terms that were enriched with the genes whose gene bodies harbor the differentially-edited A-to-I editing sites between tumor and matched normal tissues. The terms found in at least two pathologies of glioma were shown.

in cancer progression[19]. Interestingly, grade 2 oligodendroglioma is unique among the gliomas we studied in that downregulation of Alu RNA is accompanied by downregulation of other TEs (Fig. 3b). In other pathologies, non-*Alu* transposable elements do not show an overall bias of differential expression toward up- or downregulation. One possible mechanism behind the downregulation of TE-derived RNA is histone modifications. Histone 3 lysine 9 trimethylation (H3K9me3), a repressive histone mark was reported to be increased as a stress response, resulting in downregulation of RNA of transposable elements in mouse brain[42]. As H3K9me3 was reported to be perturbed differentially depending on glioma pathology[43], it will be interesting to test the direct effect of H3K9me3 on the expression of transposable elements including Alu element in different types of glioma.

Among *Alu*-associated regulatory processes, two post-transcriptional processes, A-to-I editing and backsplicing generating circular RNA are known to be abundant in brain tissue[22,23]. Many previous studies have shown that these processes are affected in gliomas. For example, A-to-I editing is significantly altered, usually reduced in glioma[25]. And there had been reports highlighting that alterations in specific RNA-editing sites can contribute to tumor progression and classification of

molecular signatures or grades in GBM[26,44–47]. Moreover, a recent study comparing relative genome-wide RNA-editing levels among genetic subgroups of glioma showed that the RNA-editing signature can be used for the prediction of *isocitrate dehydrogenase* (IDH) mutation and chromosome 1p/19q-codeletion status in gliomas[48]. Regarding circular RNA, emerging evidence is accumulating on aberrant circular RNAs expression and its oncological function in gliomas[49–51]. Studies have implicated circular RNAs in the proliferation, invasion, and angiogenesis of gliomas through cancer-associated signaling pathways[52]. However, these studies mostly focused on individual sites or genes limiting their biological implications in terms of *Alu*-associated processes in brain cancer. Our results of global decreases of both A-to-I editing and circular RNAs suggest that there are perturbations affecting the molecular mechanism of these processes in glioma, beyond individual sites or genes. We also showed that these changes, especially of A-to-I editing levels might be caused by the downregulation of *ADAR2*. But it should be noted that we only observed mRNA levels of *ADAR*, requiring further investigation of protein levels of ADAR enzymes as *ADAR* mRNA expression can be uncorrelated with protein levels or activity. For example, it has been reported that subcellular localization of

ADAR2 proteins is involved in the change of A-to-I editing levels in neural development[53].

A-to-I editing and circular RNA are posited to mediate *Alu* element's contribution to the evolution of the human brain. We and others have shown that A-to-I editing levels and circular RNA are abundant in neural genes that are enriched in the pathways of neurotransmission, neurogenesis, and synaptogenesis[10,22,23,54,55]. *Alu* embedded in neural genes may serve as a mechanism to expand diversities in their functions and regulations that underlie remarkable complexities in the human nervous systems[20,56]. The evolutionary benefits that *Alu* offers in human brain development, however, may turn into *Alu* element-specific adverse effects in pathological conditions in the nervous systems. Along with this idea, the Alu neurodegeneration hypothesis[24] proposes *Alu* as a double-edged sword, whereby beneficial Alu-related processes also have the potential to disrupt mitochondrial homeostasis in neurodegenerative disorders including Alzheimer's disease. We propose that dysregulation of A-to-I editing and circular RNAs observed in higher grades of gliomas, affecting many biological processes including glutamate signaling, are also a manifestation of the double-edged effect of *Alu* in glioma. From this evolutionary perspective, recent studies offer a clue as to how the beneficial effects of *Alu* might be turned into a tumor-promoting factor. For example, Venkataramani et al.[57] reported that neuronal activities associated with glutamate synaptic connections contribute to the progression of glioma. Here, *Alu* may enhance neuronal activity through A-to-I editing, an evolutionary advantage, but it also makes perturbation of A-to-I editing enzymes in glioma a favorable environment for glioma progression. The experimental investigation will be essential to test the causality of Alu elements on gliomagenesis through the changes of A-to-I editing levels or circular RNA levels in future studies.

As *Alu* involves both A-to-I editing and the generation of circular RNA, it is intriguing to see whether these two *Alu*-associated post-transcriptional processes are influencing each other. The relation between circular RNA and A-to-I editing is unclear. Although some studies showed that circular RNA expression has a negative correlation with expression of ADAR1 in human cell lines[38,55], a recent study did not find a global correlation between the two processes in mouse tissues[58]. We observed little correlations between A-to-I editing levels and circular RNA levels in general although some genes can have interdependent changes between A-to-I editing and circular RNA formation through A-to-I editing in the flanking introns of circular RNA. However, our analyses were performed at a gene level, allowing only a limited understanding of interdependence between the two processes. Also, our results showed the different patterns between Alu RNA expression and A-to-I editing level or circular RNA formation in O2: the Alu expression levels are most strongly changed across all the glioma subtypes, whereas both the A-to-I editing and circular RNA are changed most slightly. We speculate that the molecular mechanisms between Alu expression change and the perturbation of A-to-I editing or circular RNA are distinct. The strong downregulation of Alu expression in O2 is possibly caused by epigenetic mechanisms, such as a repressive histone mark of H3K9me3 as mentioned earlier, whereas the least effect on A-to-I editing and circular RNA in O2 might be due to the little change of ADAR2 expression. Further studies are required to understand the molecular mechanisms behind the uniqueness of oligodendroglioma and to uncover the relations between A-to-I editing and circular RNA.

## Methods

**Patient tissues**. Surgical specimens and clinical information were obtained from glioma patients who underwent surgery at Seoul National University Hospital,

Seoul, South Korea. Informed consent was obtained from all patients for the usage of samples. A total of 42 patients with a matched pair of tumor and normal samples were enrolled. Normal brain tissues were obtained when the surgical approach to the tumor involves brain areas without evidence of microscopical involvement of the tumor. The final diagnosis was rendered using the most recent update of cIMPACT-NOW guidelines[27,28]. After performing stranded RNA-seq, we removed one patient for further analysis as its tumor sample has lower sequencing quality based on the number of counted reads for gene expression measurement (lower than 10% of total sequencing reads). This study was performed under the approval of the Institutional Review Board of Seoul National University Hospital, Seoul, South Korea (IRB approval No., H-1404-056-572).

**Generation and processing of RNA-seq**. RNA-seq libraries were generated with total RNA extracted from tissues by using the commercial kit of Illumina TruSeq Stranded Total RNA LT Sample Prep Kit. We performed paired-end sequencing, generating a 101 nucleotides sequencing read for each end. Sequencing reads were aligned to the human reference genome by the STAR software (version: 2.6.0c)[59], using the primary assembly and gene annotation obtained from GENCODE (GRCh38 'primary_assembly' and 'comprehensive gene annotation (regions: CHR)' version 27). The potential PCR duplicates were marked by the Picard MarkDuplicates (version: 2.6.0). The numbers of reads assigned to gene bodies were counted in a strand-specific way by using FeatureCounts (version: subread 1.6.2) with the command-line options of '-g gene_id -C -s2 -p'[60].

**Gene expression analyses**. t-SNE was performed using the R package Rtsne (version: 0.15)[61], with TPM (transcripts per million reads) values. Statistical tests for differential expression between tumor and matched normal tissues were done by using DESeq2[62] with a regression model that has patient information as a covariate.

**RNA-editing call for a sample with RNA-seq**. We developed a computational pipeline to call RNA-editing sites from strand-specific RNA-seq. First, the following reads were filtered to reduce potential technical errors in the identification of RNA-editing sites: (i) reads were suspected as PCR duplication, which was determined by Picard MarkDuplicates, (ii) reads were mapped at multiple loci, (iii) reads had more than or equal to 5 nucleotides that were clipped by aligner, (iv) reads had more than 10 nucleotides composing homopolymers, where a homopolyer was defined as 4 or more contiguous same nucleotides, (v) reads had any insertion or deletion. Second, mismatches relative to the reference genome were identified. Third, for every genomic site that has reads with any mismatch whose sequencing quality is greater than 20 (in a Phred score scale), the sequencing reads were counted separately for the reference sequence or mismatches. Fourth, filter sites if mismatches were found closer to either 5' or 3' sequencing ends (<=5 nucleotides) in more than half the number of reads with mismatches. Also, if there were multiple alternative alleles, keep the sites only when there is a major alternative allele such that the major allele has at least five sequencing reads and more than two times of sequencing reads than any other alternative alleles. Finally, if a site has more than or equal to 3 sequencing reads for an alternative allele and does not overlap with SNP (dbSNP 150 common except for variants discovered using a cDNA template), it was called as a potential RNA-editing site whose type is determined by considering the strand information of RNA-seq.

**Identification of differentially-edited sites between tumor and matched normal tissue**. The sites shared by all the patients for a given pathology were statistically tested for whether they showed different A-to-I editing levels between tumor and matched normal tissues. Specifically, beta-binomial regression was performed for a given site while controlling patient-specific effect by adding patient information as a covariate in the regression. The fitting of the regression model was done with R functions in Redit package[63]. The A-to-I editing level for a given sample was defined as the ratio of inosine-supporting read counts relative to the adenosine or inosine-supporting read counts. Multiple test correction was done by using false discovery rate (FDR). The sites whose FDR-adjusted p-value is less than 0.05 were called as differentially-edited sites.

**Identification and analysis of circular RNA**. Circular RNAs were identified using CIRCexplorer2 (version: 2.3.8)[64] with STAR (version: 2.6.0c)[59] options of '--clip3pNbases 1 --chimSegmentMin 10'. We focused on exonic circular RNAs based on the gene annotation from GENCODE (version 27). In order to compare circular RNA expression rates (the degree of backsplicing relative to total splicing) across the samples, we performed beta-binomal regressions for every annotated genes, where the number of back-splicing junction reads identified by CIRCexplorer2 was modelled to follow binomial distribution with a total number of both back-splicing and linear junction reads as a parameter. Statistical significances of differences between tumor and matched normal samples were calculated while controlling patient-specific effect with a covariate in the regression if a gene body has more than three back-splicing junction reads in any comparing samples.

**Alu element expression analysis**. We measured expression levels of Alu elements using TEtranscripts (version: 2.1.4)[65] with STAR (version: 2.6.0c)[59] options of '--clip3pNbases 1 --winAnchorMultimapNmax 100 --outFilterMultimapNmax 100' and the gene annotation from GENCODE (version 27). Briefly, TEtranscripts is a software that utilizes both uniquely and ambiguously mapped reads to quantify RNA expression levels of transposable elements from RNA-seq. Statistical tests for differential expression between tumor and matched normal tissues were done by using DESeq2[62] with a regression model that has patient information as a covariate.

**Other bioinformatic analyses**. Repeat annotation was according to the UCSC RMSK track. The intersection analysis and Gene Ontology (GO) analysis in the Fig. 4 were done for the genes whose mean TPM values were greater than 1 in either tumor or normal tissues per a pathology. In GO analysis, we used NCBI RefSeq Gene annotation and the R package, GOstats[66]. The terms in the biological process whose FDR-adjusted $p$-value is less than 0.1 were called as over-represented terms.

**Statistics and Reproducibility**. Statistical analysis was performed using the statistical computing software R (version 3.6). The number of samples in a statistical analysis is specified in the result of the statistical test or the corresponding figure legend. For main figures, see the followings. Figure 1d: Wilcoxon text (paired), $N = 6, 9, 4, 5, 15$ patients for O2, O3, A2, A3, GBM, respectively; Fig. 2a: Fisher exact test, $N = 58,288$ genes; Fig. 2b: two-sided paired $t$ test, $N = 41$ patients; Figs. 3b and 4a: two-sided Wald test implemented in a R package DESeq2, $N = 12, 18, 8, 10, 4, 30$ tissues for O2, O3, A2, A3, A4, GBM, respectively; Fig. 4b: Fisher exact test, $N = 22725$ genes; Fig. 4c: two-sided Kolmogorov-Smirnov test, $N = 1578$ A-to-I editing sites.

**Ethics approval and consent to participate**. This study involving human tissue and cells was approved by the Ethics Committee of the Seoul National University Hospital (IRB No. H-1404-056-572). All tissue and data were anonymized. This study was performed in accordance with the Declaration of Helsinki.

**Reporting summary**. Further information on research design is available in the Nature Research Reporting Summary linked to this article.

## Data availability
RNA-seq raw data can be accessed through NCBI GEO (GSE165595). Also, source data underlying the following main figures is presented in the Supplementary Data: Fig. 1a (Supplementary Data 6), Figs. 1b, d and 2b (Supplementary Data 7), Fig. 2a (Supplementary Data 8), Fig. 3a, c (Supplementary Data 9) and Fig. 4a (Supplementary Data 10).

## Code availability
The custom R codes for the bioinformatic analyses are available from the corresponding authors upon reasonable request.

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

## Acknowledgements

We thank Dr. Daniel R. Weinberger (Lieber Institute for Brain Development and Johns Hopkins University) for discussion and advising manuscript. This study was supported by Seoul National University Hospital Research Fund (2620160010) and the Basic Science Research Program (NRF-2019R1A2C2005144) through the National Research Foundation of Korea (NRF) funded by the Ministry of Science & ICT of Republic of Korea.

## Author contributions

T.H. and C.K.P. designed research; T.H. developed computational pipelines and performed statistical analyses; S.K., T.C., H.J.Y., K.M.K. and H.K. contributed to sample preparation and data generation; J.-K.W. and S.H.P. confirmed the genetic classification of patient samples; JHS contributed analytic resources and discussed the results; T.H. and C.K.P. wrote the manuscript.

## Competing interests

The authors declare no competing interests.
