## [Transparent Peer Review File · Communications Biology]

Reviewers' comments:

Reviewer #1 (Remarks to the Author):

The authors used strand specific RNA-seq analysis of 41 matched normal-tumor sample pairs from glioma patients to “investigate the relevance of Alu to the gliomagenesis ...”. The approach used seems technically sound and the results appear to be robust. The role of transposable elements like Alu, long dismissed as junk DNA with no function beyond their own replication, in a variety of biological process is increasingly being recognized. Thus, work of this kind is important and worthy of publication in principle. I have conceptual and technical concerns that I would like to see addressed before the paper is published.

My main conceptual concern is whether the results support the conclusion that “Alu is associated with glioma development based on RNA expression and post-transcriptional regulations”. The aim of the study as stated in the abstract is “to investigate the relevance of Alu to the gliomagenesis ...”. The authors convincingly show that Alu elements are associated with reduced A-to-I RNA editing and circular RNA in gliomas. What is not clear is how the authors can distinguish an active, and potentially causal, role for Alu-associated RNA editing events and circular DNA formation in gliomagenesis as opposed to the possibility of Alu elements being simply a passive marker of decreased ADAR2 expression in gliomas.

My main technical concern relates to the computational pipeline developed to call RNA editing sites from strand-specific RNA-seq. The pipeline discards reads that are mapped to multiple locations. This is a standard approach for RNA-seq but could introduce bias when analyzing a highly repetitive element like Alu. There are several approaches available to rescue reads mapped to multiple locations, which are often employed in *-seq analyses of transposable elements. In fact, the authors use one such method (TEtranscripts) to measure Alu expression. It is not clear why multi-mapped reads were discarded for the RNA editing calls but included for the Alu expression analysis.

Reviewer #2 (Remarks to the Author):

In this manuscript, Hwang and colleagues systematically evaluated A-to-I editing and circRNA expression between neurotypical brain tissues and diverse types of gliomas, and tried to investigate the possible role of Alu elements during brain tumor pathogenesis. Although authors observed that there were decrease in both A-to-I editing and circRNA expression in gliomas, current evidence shown in this manuscript was not sufficient to decipher any convincing conclusion about the Alu-associated post-transcriptional regulation mechanisms. Some concerns need to be carefully addressed as listed below.

1. It would be more appropriate to use the wilcoxon signed-rank test rather than paired t-test in the comparisons, especially for non-normal distribution data.
2. Although a total of 42 patients were included in this study, patients belong to six different grades of gliomas, and most types of gliomas only have less than 10 patients. The number of patients in each glioma might be not sufficient to obtain enough statistical power.
3. As previous studies (Zhang et al., 2014, Cell; Liang et al., 2014, gene) showed that back-splicing can be facilitate by Alu elements in the flanking introns of circRNAs, which can form inverted complementary sequences to bring back-splice sites together. In this manuscript, authors only checked the overlap between A-to-I editing sites and circRNAs occurred in the same genes. Are these A-to-I editing sites located in Alu elements in the flanking introns of corresponding circRNAs? If are, do these editing sites impair the pairing capacity of Alu elements to impact the biogenesis of circRNAs.
4. Authors only showed that both the expression of ADAR2 and A-to-I editing levels were decreased in gliomas. There is no any direct evidence to show the decrease of ADAR2 expression is the cause leading to the decrease of A-to-I editing. According to evidence listed in this manuscript, we cannot

obtain any conclusion about the molecular mechanism of ADAR2 on Alu-associated post-transcriptional regulation.

Reviewer #3 (Remarks to the Author):

Brief summary of the manuscript

This paper performed strand-specific RNA-seq for 41 pairs of neurotypical brain tissue samples and samples of diverse gliomas and confirmed that A-to-I editing and circular RNAs are significantly associated with Alu.

Overall impression of the work

The impressive side of this work is that it performed RNA-seq for 41 pairs of neurotypical brain tissue samples and samples of diverse gliomas with the corresponding normal samples; the sequencing data is precious.

However, the overall analysis seems quite limited. The mechanisms, especially the mechanism between A-to-I editing, circular RNA and Alu RNA expression, are underexplored and explained. Specific comments, with recommendations for addressing each comment

Major:

- 1) All the subtypes of brain cancers studied do not have a very clear pattern (Figure 1B). Due to the small sample size in the view of statistics (I understand it is difficult to collect more samples), the result is not very conclusive (Figure 1D). It may also be useful also to check other clinical information (e.g., patient gender, age) to see if these variables also have some effect on or introduce some biases to the resulting pattern.
- 2) It would be interesting to discuss more on the opposite directions of A-to-I editing changes between A(A2, A3) and O(O3,O2) subtypes.
- 3) Related to 1), the authors emphasized a lot on O2 for A-to-I editing, but the overall pattern of O2 is quite dispersed (Figure 1D). It is hard to say whether the 3 points above the median line are due to difference introduced by clinical variables (e.g., age,) or not.
- 4) The mechanism between A-to-I editing, circular RNA and Alu RNA expression is poorly explained. For example, for O2, the Alu expression levels are most strongly changed across all the cancer subtypes, but both the A-to-I editing and circular RNA are changed most slightly across all the cancer subtypes. The logic seems quite contrary to expectation. But if this observation is true, it is also intriguing. In any case, explanations/hypotheses should be given.
- 5) ADAR2 seems a useful explanation for the change of Alu expression. But it is good to discuss and distinguish between ADAR2 gene expression and protein expression levels.

Minor:

- 1) It is good to have the same color to indicate the same cancer subtype across different figures. Orange indicated GBM in Figure 1B but green indicated GBM in Figure 1D.
- 2) neurotypical -> neurotypical

Response to Reviewers' comments

COMMSBIO-21-1455

Genome-wide perturbations of *Alu* expression and *Alu*-associated post-transcriptional regulations find a uniqueness of oligodendroglioma in glioma

Reviewer #1 (Remarks to the Author)

The authors used strand specific RNA-seq analysis of 41 matched normal-tumor sample pairs from glioma patients to “investigate the relevance of *Alu* to the gliomagenesis ...”. The approach used seems technically sound and the results appear to be robust. The role of transposable elements like *Alu*, long dismissed as junk DNA with no function beyond their own replication, in a variety of biological process is increasingly being recognized. Thus, work of this kind is important and worthy of publication in principle. I have conceptual and technical concerns that I would like to see addressed before the paper is published.

We thank the reviewer for appreciating our efforts on analyzing matched glioma samples and our perspective of glioma on *Alu* element.

My main conceptual concern is whether the results support the conclusion that “*Alu* is associated with glioma development based on RNA expression and post-transcriptional regulations”. The aim of the study as stated in the abstract is “to investigate the relevance of *Alu* to the gliomagenesis ...”. The authors convincingly show that *Alu* elements are associated with reduced A-to-I RNA editing and circular RNA in gliomas. What is not clear is how the authors can distinguish an active, and potentially causal, role for *Alu*-associated RNA editing events and circular DNA formation in gliomagenesis as opposed to the possibility of *Alu* elements being simply a passive marker of decreased ADAR2 expression in gliomas.

We agree with the reviewer’s point. Although our results showed that some gliomas have decreased A-to-I editing levels and circular RNA levels in *Alu* regions, we are not able to determine whether these molecular changes cause gliomagenesis. We admit this limitation in our revised manuscript (line 301).

Experimental investigation will be essential to test causality of Alu elements on gliomagenesis through the changes of A-to-I editing levels or circular RNA levels in future studies.

My main technical concern relates to the computational pipeline developed to call RNA editing sites from strand-specific RNA-seq. The pipeline discards reads that are mapped to multiple locations. This is a standard approach for RNA-seq but could introduce bias when analyzing a highly repetitive element like *Alu*. There are several approaches available to rescue reads mapped to multiple locations, which are often employed in *-seq analyses of transposable elements. In fact, the authors use one such method (TEtranscripts) to measure *Alu* expression. It is not clear why multi-mapped reads were discarded for the RNA editing calls but included for the *Alu* expression analysis.

We thank the reviewer for pointing out an important technical issue in RNA editing call.

Although it is conceptually better to take advantage of multi-mapped reads in RNA editing call, it is common to filter out those reads or to take a simple approach of random assignment in RNA editing pipeline due to high false positive rates that could be resulted from alignment ambiguity. For an example, Eli Eisenberg group recently proposed a computational measure quantifying A-to-I editing degree for a given sample in the Nature Methods paper (*Nature Methods* 16, no. 11 (November 2019): 1131–38). In their analysis of A-to-I editing in Alu region, they filtered out secondary alignments for multi-mapped reads, equivalent to the random assignment of multi-mapped reads. For another example, Jin Billy Li group published a Nature Methods (2012) paper describing a computational pipe to identify RNA editing sites from RNA-seq (*Nature Methods* 9, no. 6 (June 2012): 579–81). They argued that their pipeline used “meticulous” analyses of genomic DNA and RNA sequences, where only with uniquely mapped reads for both Alu and non-Alu regions were used in their computational pipeline. Therefore, we took a conservative approach of disregarding multi-mapped reads and we don’t think that additional consideration of multi-mapped reads change our results.

Reviewer #2 (Remarks to the Author)

In this manuscript, Hwang and colleagues systematically evaluated A-to-I editing and circRNA expression between neurotypical brain tissues and diverse types of gliomas, and tried to investigate the possible role of Alu elements during brain tumor pathogenesis. Although authors observed that there were decrease in both A-to-I editing and circRNA expression in gliomas, current evidence shown in this manuscript was not sufficient to decipher any convincing conclusion about the Alu-associated post-transcriptional regulation mechanisms. Some concerns need to be carefully addressed as listed below.

1. It would be more appropriate to use the wilcoxon signed-rank test rather than paired t-test in the comparisons, especially for non-normal distribution data.

We thank the reviewer for this suggestion. We added the results of Wilcoxon signed-rank test for all the statistical analyses previously done by paired t-test in the revised manuscript: line 119, lines 146-151, line 170, lines 176-181, line 194, line 623. No statistical significance was changed.

2. Although a total of 42 patients were included in this study, patients belong to six different grades of gliomas, and most types of gliomas only have less than 10 patients. The number of patients in each glioma might be not sufficient to obtain enough statistical power.

We agree with the reviewer’s concern. We made our best efforts to collect a larger number of samples, but it was difficult for us to conduct a study with a large cohort of patients which needs collaboration of multiple institutes especially at this time of COVID-19 pandemic. It is totally possible to lose statistical power not being able to capture important molecular features with weak or moderate effects. We discuss this limitation in the discussion of the revised manuscript as shown below (line 245).

We found that Alu-associated molecular processes including Alu RNA expression, A-to-I editing levels and circular RNA formation are perturbed in gliomas, whereas it is possible that we might not capture moderate perturbations due to the limited number of patients in each glioma.

3. As previous studies (Zhang et al., 2014, Cell; Liang et al., 2014, gene) showed that back-splicing can be facilitated by Alu elements in the flanking introns of circRNAs, which can form inverted complementary sequences to bring back-splice sites together. In this manuscript, authors only checked the overlap between A-to-I editing sites and circRNAs occurred in the same genes. Are these A-to-I editing sites located in Alu elements in the flanking introns of corresponding circRNAs? If are, do these editing sites impair the pairing capacity of Alu elements to impact the biogenesis of circRNAs.

According to the reviewer's suggestion, we performed the detail analysis of positional overlap between A-to-I editing sites and circular RNAs. First, we evaluated the positional overlap between Alu-associated A-to-I editing sites and the flanking introns of circular RNAs for the 91 genes that have both significant A-to-I editing sites and significant circular RNA expression rates across pathologies. Note that the numbers in Figure 4B were changed in the revision as we updated the gene annotation from NCBI refSeq to Gencode in this analysis, as follows:

In 73% (66 genes) among the 91 genes, the perturbed A-to-I editing sites were observed in the flanking introns of any circular RNAs in the same genes. Second, we found that no A-to-I editing sites are found in the flanking intron regions in the 395 genes that have significant A-to-I editing sites but no changes in circular RNA rates. This is in contrast to the first result of the 91 genes (Fisher exact test p -value $< 10^{-15}$), suggesting the potential effect of the A-to-I editing sites on the formation of circular RNA when located in the flanking regions of circular RNA, possibly by perturbing the pairing capacity of Alu elements. A similar speculation was proposed in a previous study [Andranik Ivanov et al., *Cell Reports* 10, no. 2 (January 13, 2015): 170–77]. They showed that knockdown of the double-strand RNA-editing enzyme ADAR1/2 significantly and specifically upregulated circRNA expression. But the detail analysis at the level of individual circular RNAs should be necessary in the future study to confirm this hypothesis. We mentioned these results in the discussion of the revised manuscript: line 281, line 319.

We also tested whether Alu-associated A-to-I editing sites are located in the flanking introns of circular RNAs, which may affect the formation of circular RNA [36]. We found that in 73% (66 genes) among the 91 genes, the perturbed A-to-I editing sites were observed in the flanking introns of any circular RNAs in the same genes.

We observed little correlations between A-to-I editing levels and circular RNA levels in general although some genes can have interdependent changes between A-to-I editing and circular RNA

formation through A-to-I editing in the flanking introns of circular RNA. However, our analyses were performed at a gene level, allowing only a limited understanding of interdependence between the two processes.

4. Authors only showed that both the expression of ADAR2 and A-to-I editing levels were decreased in gliomas. There is no any direct evidence to show the decrease of ADAR2 expression is the cause leading to the decrease of A-to-I editing. According to evidence listed in this manuscript, we cannot obtain any conclusion about the molecular mechanism of ADAR2 on Alu-associated post-transcriptional regulation.

We agree that RNA information only provide the indirect evidence of ADAR2' involvement of Alu-mediated post-transcriptional regulations such as A-to-I editing. However, it is well known that ADAR2 is the main enzyme responsible for A-to-I editing in brain tissues. And there are multiple literatures showing that ADAR2 RNA expression levels affect A-to-I editing levels. For example, Peng et al. (Neuron 49, Issue 5, March 2006, Pages 719-733) showed that reduced ADAR2 mRNA expression positively affects a famous A-to-I editing site, GluR2 Q/R site editing level in individual vulnerable neurons: the figure below was the unmodified figure 1I from this paper. Also, the Gallo group showed that down-regulation of ADAR2 by siRNA decreased A-to-I editing level in human brain cancer cell line (U118) in their paper published in Oncogene (2013, volume 32, 998-1009). But we still acknowledge that A-to-I editing changes can be affected by multiple mechanisms at a protein level. A previous study showed that subcellular localization of ADAR2 protein affects A-to-I editing change in brain development (J Cell Sci (2017) 130 (4): 745–753). Therefore, we discuss the limitation of our results in the revised manuscript (line 281) as shown below:

But it should be noted that we only observed mRNA levels of ADAR, requiring further investigation of protein levels of ADAR enzymes. It is possible that mRNA expression is uncorrelated with protein levels or activity. For an example, it has been reported that subcellular localization of ADAR2 proteins is involved in the change of A-to-I editing levels in neural development [49].

Figure 1I excerpted (no modification) from Neuron Volume 49, Issue 5, 2 March 2006, Pages 719-733. The increasing levels of ADAR2 mRNA causes higher A-to-I editing levels.

Reviewer #3 (Remarks to the Author)

Brief summary of the manuscript

This paper performed strand-specific RNA-seq for 41 pairs of neurotypical brain tissue samples and samples of diverse gliomas and confirmed that A-to-I editing and circular RNAs are significantly associated with Alu.

Overall impression of the work

The impressive side of this work is that it performed RNA-seq for 41 pairs of neurotypical brain tissue samples and samples of diverse gliomas with the corresponding normal samples; the sequencing data is precious.

We thank the reviewer for appreciating our efforts on analyzing matched glioma samples by using RNA-seq.

However, the overall analysis seems quite limited. The mechanisms, especially the mechanism between A-to-I editing, circular RNA and Alu RNA expression, are underexplored and explained. Specific comments, with recommendations for addressing each comment

Major:

1) All the subtypes of brain cancers studied do not have a very clear pattern (Figure 1B). Due to the small sample size in the view of statistics (I understand it is difficult to collect more samples), the result is not very conclusive (Figure 1D). It may also be useful also to check other clinical information (e.g., patient gender, age) to see if these variables also have some effect on or introduce some biases to the resulting pattern.

We thank the reviewer for constructive suggestion. We checked if either patient gender or age affects the A-to-I editing in our samples. According to the reviewer's concern in the Figure 1D, we looked into the correlations between gender or age and alu editing index (AEI). We did not see any correlation as shown below (left: gender, right: age). In our view, it is unlikely that other clinical information, at least patient gender and age, introduces biases on our observed patterns of A-to-I editing levels.

observed: p-value=0.394 by Welch Two Sample t-test and p-value=0.6659 by Wilcoxon rank sum test.	
--	--

2) It would be interesting to discuss more on the opposite directions of A-to-I editing changes between A(A2, A3) and O(O3,O2) subtypes.

We want to be cautious about this pattern as mentioned in the result section: *this trend might be due to differing sample sizes per pathology and should be interpreted cautiously.*

But we agree with the reviewer that it would be interesting to think about the implication of the pattern. A recent single cell RNA-seq study on IDH-mutant gliomas, Venteicher et al. (*Science* (2017) 355, no. 6332) uncovered that IDH mutant astrocytoma (IDH-A) and IDH mutant oligodendroglioma (IDH-O) have different tumor micro-environment including the infiltration of microglia or microphage: IDH-A have a higher proportion of microglia/macrophage than do IDH-O. Also, they showed that the different grades of IDH-A have different degrees of microglia/microphages. We think that this difference can affect the opposite directions of A-to-I editing changes observed in our glioma samples as the changes in cell composition can affect A-to-I editing patterns (Scientific reports (2017) 7: 43421). It might be possible that A-to-I editing changes in IDH-A are dominantly affected by differing cell compositions, whereas IDH-O has different A-to-I editing levels due to decreasing levels of ADAR enzymes in relative homogeneous cell population.

3) Related to 1), the authors emphasized a lot on O2 for A-to-I editing, but the overall pattern of O2 is quite dispersed (Figure 1D). It is hard to say whether the 3 points above the median line are due to difference introduced by clinical variables (e.g., age,) or not.

It is possible that the 3 points in O2 in Figure 1D are outliers. However, as shown in the response of (1) raised by this reviewer, there is no statistical evidence that other clinical variables, at least age and sex affect our results. To our best with the current dataset and the available information, we think that it is reasonable to conclude that O2 has no significant changes.

4) The mechanism between A-to-I editing, circular RNA and Alu RNA expression is poorly explained. For example, for O2, the Alu expression levels are most strongly changed across all the cancer subtypes, but both the A-to-I editing and circular RNA are changed most slightly across all the cancer subtypes. The logic seems quite contrary to expectation. But if this observation is true, it is also intriguing. In any case, explanations/hypotheses should be given.

We speculate that the molecular mechanisms between Alu expression change and the perturbation of A-to-I editing or circular RNA are uncoupled. As we mentioned in the discussion, strong down-regulation of Alu expression in O2 is possibly caused by a repressive histone mark of H3K9me3. Hunter et al. (PNAS 2012,109:17657–62) reported H3K9me3 to be increased as a stress response, resulting in down-regulation of RNA of transposable elements in mouse brain. Also, Venneti et al. (J Neuropathol Exp Neurol. 2013;72:298–306) found that

H3K9me3 was perturbed differentially between IDH-mutant oligodendroglioma and high-grade astrocytoma. It will be interesting to test the direct effect of H3K9me3 on the expression of transposable elements including Alu element in glioma. But, the changes of A-to-I editing and circular RNA in glioma may be affected by the perturbation of activities of responsible enzymes such as ADAR2. Therefore, the opposite degrees of changes in Alu RNA expression and two Alu-associated post-transcriptional regulations observed in O2 can be understood with distinct molecular mechanisms. We added this discussion in the revised manuscript (line 312).

our results showed the different patterns between Alu RNA expression and A-to-I editing level or circular RNA formation in O2: the Alu expression levels are most strongly changed across all the glioma subtypes, whereas both the A-to-I editing and circular RNA are changed most slightly. We speculate that the molecular mechanisms between Alu expression change and the perturbation of A-to-I editing or circular RNA are distinct. The strong down-regulation of Alu expression in O2 is possibly caused by epigenetic mechanisms such as a repressive histone mark of H3K9me3 as mentioned earlier, whereas the least effect on A-to-I editing and circular RNA in O2 might be due to the little change of ADAR2 expression. Further studies are required to understand the molecular mechanisms behind the uniqueness of oligodendroglioma and to uncover the relations between A-to-I editing and circular RNA.

5) ADAR2 seems a useful explanation for the change of Alu expression. But it is good to discuss and distinguish between ADAR2 gene expression and protein expression levels.

See the response to the point 4 raised by the reviewer 2.

Minor:

1) It is good to have the same color to indicate the same cancer subtype across different figures. Orange indicated GBM in Figure 1B but green indicated GBM in Figure 1D.

We fixed the Figure 1B in the revised manuscript to have consistent colors with the other figures.

2) neurotypicial -> neurotypical

We fixed it.

REVIEWERS' COMMENTS:

Reviewer #1 (Remarks to the Author):

I am satisfied with the authors' responses to my comments and recommend that the manuscript be accepted for publication.

Reviewer #2 (Remarks to the Author):

In the revised manuscript, although Hwang and colleagues made lots of efforts to refine their analyses, they did not reveal too many new biological mechanisms yet. They only found some molecular associations such as A-to-I editing and circRNA expression, A-to-I editing and ADAR2 expression, or Alu RNA expression and Pol III expression, rather than direct evidence or causal effects. Listing them as limitations cannot improve the quality of this manuscript, and more importantly, this manuscript did not have enough novelty.

Another concern is that there is no logic in this manuscript. Although both A-to-I editing and circRNA expression belong to Alu-associated post-transcriptional processes, they are independent to the expression of Pol III-transcribed Alu elements. This manuscript included too many unrelated analyses and only touched the surface. The topic in this manuscript needs to be focus if they want to publish.

Reviewer #3 (Remarks to the Author):

The authors have addressed all my comments.